# Detection of Small Ship Objects Using Anchor Boxes Cluster and Feature Pyramid Network Model for SAR Imagery

**Peng Chen [1] [ID], Ying Li [2], Hui Zhou [3],*, Bingxin Liu [1] [ID] and Peng Liu [1]**

[1]   Navigation College, Dalian Maritime University, Dalian 116026, China; chenpeng@dlmu.edu.cn (P.C.);
      gisbingxin@dlmu.edu.cn (B.L.); liupeng@dlmu.edu.cn (P.L.)
[2]   Environmental Information Institute, Dalian Maritime University, Dalian 116026, China; yldmu@126.com
[3]   School of Computer and Software, Dalian Neusoft Information University, Dalian 116023, China
*    Correspondence: zhouhui@neusoft.edu.cn; Tel.: +86-411- 8483-2287

**Abstract:** The synthetic aperture radar (SAR) has a special ability to detect objects in any climate and weather conditions. Consequently, SAR images are widely used in maritime transportation safety and fishery law enforcement for maritime object detection. Currently, deep-learning models are being extensively used for the detection of objects from images. Among them, the feature pyramid network (FPN) uses pyramids for representing semantic information regardless of the scale and has an improved accuracy of object detection. It is also suitable for the detection of multiple small ship objects in SAR images. This study aims to resolve the problems associated with small-object and multi-object ship detection in complex scenarios e.g., when a ship nears the port, by proposing a detection method based on an optimized FPN model. The feature pyramid model is first embedded in a traditional region proposal network (RPN) and mapped into a new feature space for object identification. Subsequently, the k-means clustering algorithm based on the shape similar distance (SSD) measure is used to optimize the FPN. Initial anchor boxes and tests are created using the SAR ship dataset. Experimental results show that the proposed algorithm for object detection shows an accuracy of 98.62%. Compared with Yolo, the RPN based on VGG/ResNet, FPN based on VGG/ResNet, and other models in complex scenarios, the proposed model shows a higher accuracy rate and better overall performance.

**Keywords:** SAR images; multi-target ship detection; anchor boxes clustering; feature pyramid networks

## 1. Introduction

Marine object detection has a wide range of applications in various maritime areas such as illegal fishing, oil-spill monitoring, and marine traffic management [1,2]. Currently, the main methods of identification of marine vessels include infrared, hyperspectral, optical, and radar imaging [3,4]. Unlike the first three methods, radars can actively send microwaves to capture objects and perform continuous imaging [5]. Among them, the synthetic aperture radar (SAR) is not affected by time and weather and has a wider imaging area. Furthermore, its image resolution stays constant as the distance from the observed object increases. Therefore, it has become a vital approach for marine ship object detection [6]. Conventional SAR image ship detection is mainly based on the detection of hull or wake features, typically utilizing features such as ship length, aspect ratio, contour features, ship size, and other geometric features [7]; two-dimensional (2D) comb features [8]; partial radar cross-section (RCS) density [9]; polarization characteristics [10]; and other scattering characteristics, to obtain specific

results. This method performs well in the detection of ships at sea, however, when ships are close to the ports or coral reefs, it is difficult to use statistical data to describe their scattering mechanism or extract useful characteristics. Therefore, this method has low detection accuracy in complex scenarios.

In recent years, deep neural networks (DNNs) have been used to address various problems in computer vision, including those of object detection. The mean average precision (mAP) of object detection using deep-learning models on PASCAL visual object classification (PASCAL VOC) dataset has been reported to reach a value of 0.8. The DNN solves the problems in object detection using two methods: two-stage and one-stage. The former constructs a multi-task loss function using the image classification loss and bounding box regression loss and mainly comprises two parts when training the network. The first step involves the training of the region proposal network (RPN). Feature extraction is performed on the image through a convolutional neural network (CNN), known as the backbone network, thereby creating a feature map. Typical backbone networks include the VGG, ResNet, and so on. Next, the regions of interests are generated using the RPN, and the background and object classes are created. The second step involves the training of the network for object-region detections, i.e., locating and updating the position of the region of interest, obtaining the object's position in the selected region of interest on the feature map, and obtaining the corresponding vectors passing through the fully connected layers, by using classification and regression separately, thereby locating the object and determining the class. Typical two-stage methods include R-CNN, fast R-CNN, faster R-CNN [11,12], multi-level feature pyramid network (MLFPN), neural architecture search and feature pyramid network (NAS-FPN) [13,14], etc.

The one-stage method in a manner similar to that presented above, first selects a backbone network to extract the features and creates a feature map. It subsequently uses a sliding window on the feature map to obtain the frame of interest. Each frame of interest includes 20 different types of confidence scores, and coordinate values of its bounding box (x and y-coordinates of the center, width, and height), as well as whether a confidence score is included for the object. This ensures that, after extracting the features from the entry, the results of detection are directly obtained from the frame of interest. Typical one-stage methods include Yolo V1-V3, single shot multibox detector (SSD) [15–17], etc. After the one-stage method extracts the features using the CNN, the detection results are directly obtained from the frame of interest. Examples of the aforementioned procedure can be found in SSD and Yolo, the former being highly accurate and the latter being fast at real-time detection. At present, image-detection models are widely used for ship detection in SAR images [18–21]. (Liu et al., 2017) used sea-land segmentation to obtain the ship's location of interest, and subsequently used a CNN to differentiate between the vessels [18]. (Kang et al., 2017) used a faster R-CNN for the initial detection of the ship, and then used radar constant false alarm rate (CFAR) detection to obtain the final detection result [19]. Kang et al., (2017) proposed a region-based convolutional network for ship detection with multi-layer features, which combines low-level high-resolution features and high-level semantic features to improve the detection accuracy [20]. Wang et al., (2018) used SSD to detect ships in complex backgrounds in sentinel-1 SAR images and used transfer learning to improve accuracy [21]. The first three methods mentioned above mainly focus on detection accuracy without considering environment complexity, especially near ports and shores. The latter considers the environment complexity, but not the multi-object and multi-scale characteristics of ship objects. Zhao et al., (2019) add the mutil-level information to the mutil-scale of the FPN structure and design the MLFPN network structure. Combine the same scale in different levels to improve detection accuracy [13]. This document uses neural architecture search and finds a new feature pyramid architecture in a new scalable search space that includes all cross-scale connections. It achieved accuracy and delay trade-offs over current best-target detection models [14].

Since SAR images mostly include multi-object and multi-scale ships in complex environments, accurate detection of objects of different scales is a fundamental challenge in the field of computer vision. At present, several object-detection models rely on the backbone CNN and are pre-trained on image-classification tasks to extract the feature maps of input images and use the last layer of

the feature vectors for object localization and classification. However, the last convolution layer is insufficient for dealing with bounding boxes of various scales, and the underlying positioning information is usually lost. Therefore, an optimized feature pyramid model is proposed to solve the aforementioned problems.

By contrast, the detection of multi-scale ship targets on the FPN network is directly related to the quality of candidate boxes based on anchor boxes. Therefore, to solve the aforementioned problems, this paper proposes an optimized FPN model with the ability to generate anchor boxes.

## 2. Related Work

### 2.1. Region Proposal Network (RPN) on a Backbone Network

A problem associated with ship object detection via SAR imaging is the low accuracy of multi-object ship identification in complex scenarios such as offshore ports and islands. Therefore, a more accurate object detection model is needed. The two-stage method constructs a multi-task loss function using the image classification loss and bounding box regression loss and mainly comprises two parts when training the network. Object detection models based on the frame of interest only use the topmost feature layer for prediction, examples of which include SPP-Net, Fast R-CNN, Faster R-CNN, etc., all of which use the features of the last layer of the network, as shown in Figure 1.

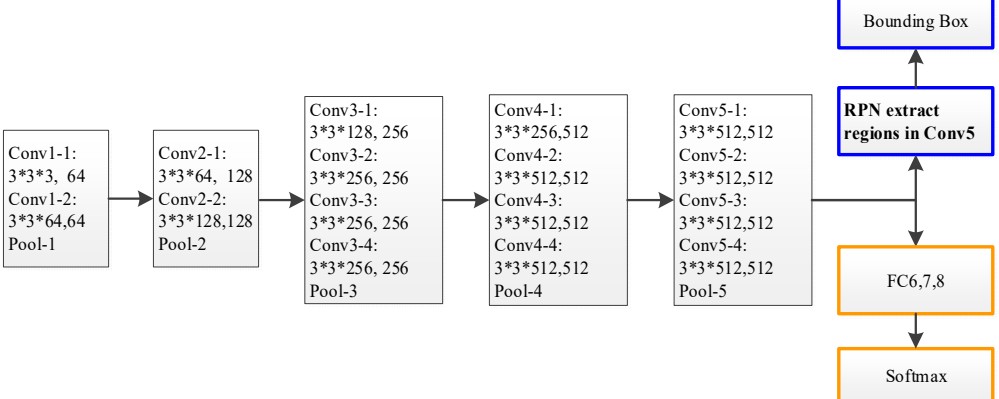

**Figure 1.** Architecture of the region proposal network (RPN) based on the backbone-network. After the original image is convolved with five layers, the last layer, Conv5, is used as a feature map to enter the positioning and detection stages.

### 2.2. Feature Pyramid Network (FPN) on Backbone

Based on the feature map extracted by the CNN, the feature semantic information of the lower layer is not as rich as the location information and the object position is more accurate, which in turn enhances the detection of small objects. Feature semantic information of higher layers is more sufficient, while the object location is less pronounced. The feature pyramid network (FPN) uses multi-scale features and top-down architecture for object detection and high-layer features with sufficient semantic information to map onto bottom-layer features with high resolution and adequate detail. The features of various layers are integrated to improve the detection of small objects.

The FPN is embedded in the RPN network, and each layer is independently predicted [22]. The image is then inserted into the pre-trained backbone network. The Conv1, Conv2, Conv3, Conv4 and Conv5 layers are the bottom-up feature maps, thereby forming the {C1, C2, C3, C4, C5} layers, and then undergoing upsampling through the top-down pathway to obtain feature maps from higher pyramid layers. {C1, C2, C3, C4, C5} are laterally connected with the upsampling results through a $1 \times 1$ convolution kernel (256 channels) to form new feature maps {P1, P2, P3, P4, P5}.

P4 = UpSampling (P5) + C4.Conv (256, (1, 1))

P3 = UpSampling (P4) + C3.Conv (256, (1, 1))
P2 = UpSampling (P3) + C2.Conv (256, (1, 1))
P1 = UpSampling (P2) + C1.Conv (256, (1, 1))

Lastly, another 3 × 3 convolution is appended to P1–P5 to eliminate the aliasing effect of upsampling. The backbone network is reorganized, as shown in Figure 2.

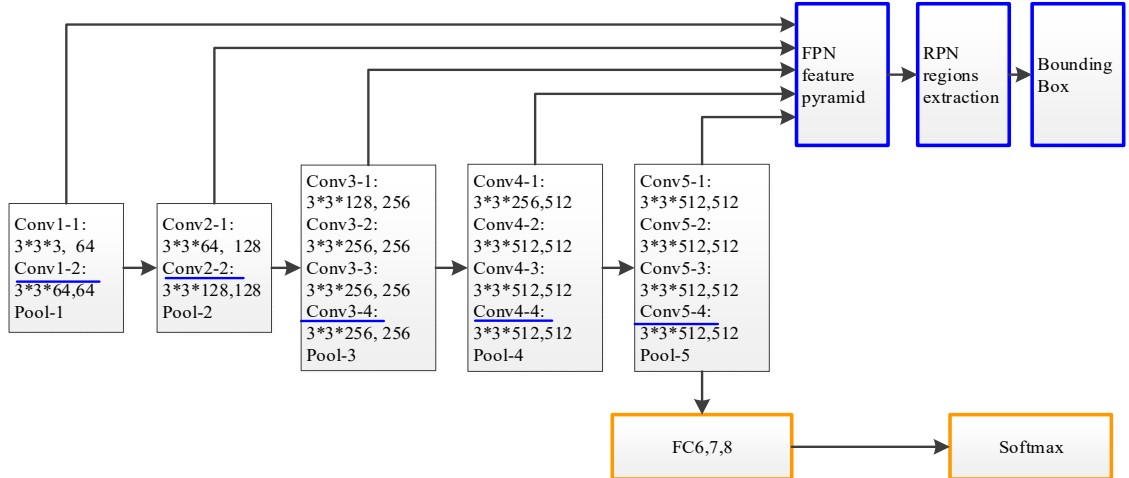

**Figure 2.** Architecture of feature pyramid network (FPN) based on the VGG. After the original image enters the convolutional neural network (CNN), features are extracted from each layer, and a mapping map is formed through upsampling and horizontal connection operations. The feature map enters the positioning and detection stage.

Three rectangular boxes (32 × 32 pixels, 64 × 64 pixels, 128 × 128 pixels) with different pixel areas are allocated on the five feature mapping layers (P1,P2,P3,P4,P5). Simultaneously, multiple aspect ratios (1:2, 1:1, 2:1) are used. These rectangular boxes are called anchor boxes. On each feature mapping layer, sliding the window with anchor boxes as a fixed area generates a large number of candidate frames, called proposals.

The training labels are assigned to the proposals based on the intersection over union (IOU) of the proposals and the actual boundary box (ground truth) ratio. If a proposal has the highest IOU for a given actual bounding box or the IOU of any actual bounding box exceeds 0.7, it is assigned a positive label, otherwise it is assigned a negative label. Four parameters are recorded at the same time, which represent the coordinates (x, y), height (h) and width (w) of the center point of the proposals. The five feature levels provide a total of proposals. Before entering the fully connected layer FC6, FPN provides 4k proposals parameters and k category parameters.

The ground truth is used as the actual value, and regression training is performed according to the proposals corresponding to the positive and negative labels, such that the proposals are closest to the ground truth. The mapping function is defined as $f$, $f(A_x, A_y, A_w, A_h) = (G'_x, G'_y, G'_w, G'_h)$, where

$$(G'_x, G'_y, G'_w, G'_h) \approx (G_x, G_y, G_w, G_h)$$
$$G'_x = A_w \cdot d_x(A) + A_x, G'_y = A_h \cdot d_y(A) + A_y$$
$$G'_h = A_h \cdot \exp(d_h(A)), G'_w = A_w \cdot \exp(d_w(A))$$

The model then has to learn the offsets $d_x(A)$, $d_y(A)$ and scaling transformations $d_w(A)$, $d_h(A)$.

## 3. Proposed Method

### 3.1. Anchor Boxes Generation Based on Shape Similar Distance (SSD)-Kmeans

Because the target ships are multi-scale, anchor boxes-based algorithms are used to generate the proposals for of the FPN map. The shapes and sizes of anchor boxes are a set of hyperparameters. In the actual SAR image, the size of the target changes considerably. The use of the anchor boxes generation mechanism in FPN leads to slower convergence of border regression and, therefore, the better initial anchor boxes need to be chosen.

The k-means algorithm is a clustering algorithm based on the distance evaluation scale to the prototype. Due to its simple and efficient characteristics, it is widely used in clustering problems. In this paper, the k-means algorithm based on the SSD clustering is used to obtain k initial anchor boxes, namely SSD-Kmeans.

The k-means algorithm involves clustering based on the evaluation scale to the prototype distance. It is widely used in clustering problems due to its simplicity and efficiency. In this paper, the k-means algorithm based on the shape similar distance (SSD) clustering is used to obtain k initial anchor boxes, namely SSD-Kmeans.

First, some ground truths need to be randomly selected as the center of the initial $k$ clusters $\mu_1, \mu_2, \ldots, \mu_k$. Subsequently, for the ground truth of each ship target $xi$, Equation (1) is used to calculate the cluster labels of the sample with SSD distance measurement.

$$label_i = \underset{1 \leq j \leq k}{\operatorname{argmin}} \, d_{\mathrm{SSD}}\big(x_i, u_j\big) \tag{1}$$

After the cluster labels of all the ground truths were obtained, each cluster center was updated using Equation (2).

$$u_j = \frac{1}{n_j} \sum_{i=1}^{nj} x_i, x_i \in C_j \tag{2}$$

Equations (1) and (2) were repeatedly calculated until the preset number of cycles or the square error expressed in Equation (3) converged to the local optimal solution.

$$E = \sum_{j=1}^{k} \sum_{i=1}^{nj} \big(d_{\mathrm{SSD}}\big(x_i, u_j\big)\big)^2, x_i \in C_j \tag{3}$$

The distance $d_{\mathrm{SSD}}$ based on shape is shown in Equations (4)–(7),

$$d_{\mathrm{SSD}}\big(\mathbf{GT}_i, \mathbf{GT}_j\big) = d_{\mathrm{ED}} \times \left[1 + \cos(\frac{d_{\mathrm{AD}}}{d_{\mathrm{MD}}} \times \frac{\pi}{2})\right] \tag{4}$$

$$d_{\mathrm{ED}}\big(\mathbf{GT}_i, \mathbf{GT}_j\big) = \sqrt{\sum_{k=1}^{4}\big(\mathbf{GT}_i - \mathbf{GT}_j\big)^2} \tag{5}$$

$$d_{\mathrm{MD}}\big(\mathbf{GT}_i, \mathbf{GT}_j\big) = \sum_{k=1}^{4}\big|\mathbf{GT}_i - \mathbf{GT}_j\big| \tag{6}$$

$$d_{\mathrm{AD}}\big(\mathbf{GT}_i, \mathbf{GT}_j\big) = \left|\sum_{k=1}^{4}\big(\mathbf{GT}_i - \mathbf{GT}_j\big)\right| \tag{7}$$

where, $d_{\mathrm{ED}}$ is the Euclidean distance, $d_{\mathrm{MD}}$ is the Manhattan distance, and $d_{\mathrm{AD}}$ is the absolute value of the vector difference. The coordinates of the ground truth are subsequently stored, and $x_{\mathrm{min}}, y_{\mathrm{min}}, w$ and $h$ of the target are recorded, respectively.

$\frac{d_{\text{AD}}}{d_{\text{MD}}}$ reflects the difference in the shape of the ground truth. The larger the value of $\frac{d_{\text{AD}}}{d_{\text{MD}}}$, the higher the similarity. The value of $\cos(\frac{d_{\text{AD}}}{d_{\text{MD}}} \times \frac{\pi}{2})$ is in the range [0, 1], when $\frac{d_{\text{AD}}}{d_{\text{MD}}} = 1$, $\cos(\frac{d_{\text{AD}}}{d_{\text{MD}}} \times \frac{\pi}{2}) = 0$, thereby indicating that the shapes of the ground truths are similar and that only the sizes are different. Otherwise, the value is in the range (0, 1), and the value of $d_{\text{SSD}}$ is in the range [$d_{\text{ED}}$, 2*$d_{\text{ED}}$]. As a distance measure, the steps to aggregate k new shapes and sizes of anchor boxes are as follows:

S1. The ground truth is randomly selected as the initial cluster center, $\mathbf{GT}_k = (x_g, y_g, w_g, h_g)$;

S2. By calculating the shape distance $d_{\text{SSD}}(\mathbf{GT}_i, \mathbf{GT}_j)$ of all other samples from the center of the k clusters, the cluster label of each sample is determined according to the closest distance;

S3. After the cluster labels are obtained for all samples, the cluster center is updated $(x'_g, y'_g, w'_g, h'_g)$ according to the mean vector;

S4. Steps S2 and S3 are calculated until the cluster center changes. The updated k cluster centers represent the corresponding new initial anchor boxes $\mathbf{A} = (x_a, y_a, w_a, h_a)$.

## 3.2. Anchor Boxes Training

The anchor boxes generated by clustering are used to replace the anchor boxes generated in FPN according to different proportions and scales. The slide on the generated feature map is used to obtain a large number of proposals. The anchor box with the largest IOU is then obtained through non-maximum suppression and regression training is performed, which in turn causes the anchor box to be closest to ground truth.

A typical pyramid can realize multi-scale feature representation and, therefore, the CNN can be integrated with image positioning, thereby combining top-down and horizontal connection to create a feature representation with strong semantics on all scales as the re-input image. Next, the input image is rebuilt, and the fully connected layers are employed for classification. The loss feedback of the reconstructed input image is then combined with the original multi-task loss function.

First, the global perception field of the fully connected layer is connected to the k convolution kernel (1*1*512) of the three fully connected layers, and the last fully connected layer corresponds to the Softmax layer. The maximum value is obtained as a probability, and the output value $p_i$ is obtained as follows:

$$p_i = \frac{e^{x_i^{fc}}}{\sum_{j=1}^{n} e^{x_j^{fc}}} \tag{8}$$

where $x_i^{fc}$ and $x_j^{fc}$ are $i$-th and $j$-th output values of the last fully connected (FC) layer, respectively.

The multi-task loss function includes the classification loss and the regression loss of locating the target box. Therefore, the loss function can be defined as follows:

$$\mathrm{L}(p, u, v_i, v_i^*) = \frac{1}{N_{cls}} \sum_i \mathrm{L}_{cls}(p_i, u_i) + \lambda \cdot \frac{1}{N_{reg}} \cdot \sum_i \mathrm{f}(u_i) \cdot \mathrm{L}_{reg}(v_i, v_i^*) \tag{9}$$

where $\mathrm{L}_{cls}(p_i, u_i)$ is the classification loss function, $\mathrm{L}_{cls}(p_i, u_i) = -\log p_i u_i$, and the probability distribution of each prediction box is $p_i = (p_0, p_1, \ldots p_k)$. Here, $k$ is the ship type, while $u$ is the predicted probability of the proposals. If the calculated proposals is a positive label, then $u_i = 1$, otherwise $u_i = 0$. The regularization parameter $\lambda$ is used to determine the weight of each task in the multi-task loss function. Here, $\mathrm{f}(u_i)$ is the indicating function, and if $[u_i \geq 1]$, then calculate $\mathrm{L}_{reg}(v_i, v_i^*)$, otherwise it is not calculated.

$L_{reg}(v_i, v_i^*)$ is the loss function for locating the target box, where $\text{smooth}_{L1}(v_i - v_i^*)$ is the smooth function of norm L1. $v_i$ and $v_i^*$ are calculated from the prediction box $(x, y, w, h)$, clustering anchor box $(x_a, y_a, w_a, h_a)$ and ground truth $(x_g, y_g, w_g, h_g)$, respectively:

$$L_{reg}(v_i, v_i^*) = \text{smooth}_{L1}(v_i - v_i^*)$$
$$\text{smooth}_{L1}(x) = \begin{cases} 0.5x^2, & \text{if } |x| < 1 \\ |x| - 0.5, & \text{otherwise} \end{cases} \tag{10}$$

$$\begin{aligned} v_x &= \frac{(x - x_a)}{w_a}, v_y = \frac{(y - y_a)}{y_a}, \\ v_w &= \log\left(\frac{w}{w_a}\right), v_h = \log\left(\frac{h}{h_a}\right), \\ v_x^* &= \frac{(x_g - x_a)}{w_a}, v_y^* = \frac{(y_g - y_a)}{y_a}, \\ v_w^* &= \log\left(\frac{w_g}{w_a}\right), v_h^* = \log\left(\frac{h_g}{h_a}\right) \end{aligned} \tag{11}$$

## 4. Experimental Process and Analysis

### 4.1. Experiment Preparation

#### 4.1.1. Dataset

China's Gaofen-3 satellite SAR and Sentinel-1 SAR datasets as the main data sources used in this study, with a total usage of 102 Gaofen-3 images and 108 Sentinel-1 SAR images [23]. The SAR dataset includes 43,819 ship data slices. Imaging models of Gaofen-3 include Strip-Map (UFS), Fine Strip-Map 1 (FSI), Full Polarization 1 (QPSI), Full Polarization 2 (QPSII), and Fine Strip-Map 2 (FSII). These five models have resolutions of 3 m, 5 m, 8 m, 25 m, and 10 m, respectively. Sentinel-1's imaging models are the stripe models (S3 and S6) with wide-field imaging. The ship target dataset is shown in Figure 3. In addition, labelling tools are used to label the vessel position and for classification. The training, verification, and testing sets, respectively, constitute 70%, 20%, and 10%. A large number of SAR images are used to train the network model in order to improve detection accuracy. The proposed workflow is illustrated in Figure 4.

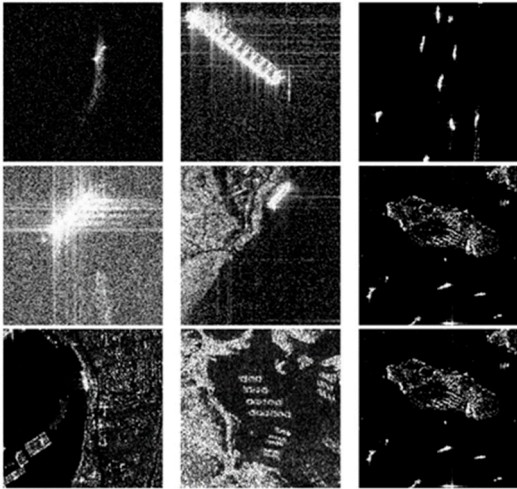

**Figure 3.** Ship target dataset in complex scenarios.

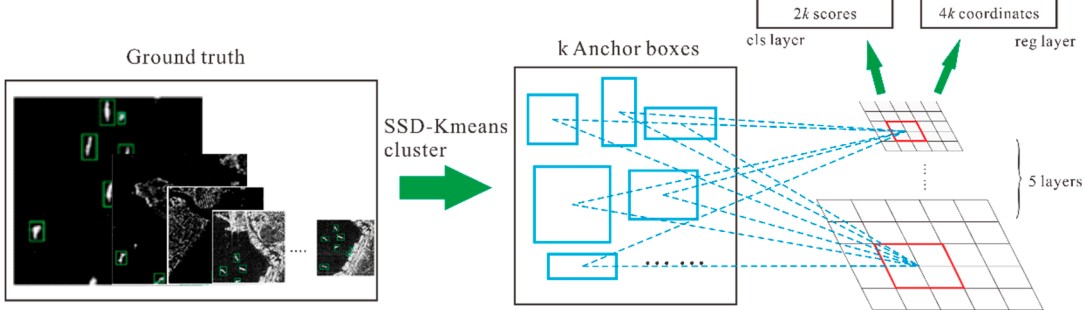

**Figure 4.** Proposed workflow. The ground truth shape of the training set is estimated and the shape similar distance (SSD)-Kmeans method is used to cluster k anchor boxes. The slide on the feature map generated by the FPN helps in obtaining a large number of proposals, and object classification and proposals regression training are performed.

### 4.1.2. Network Training

The experimental platform is Ubuntu 14.0, the graphics processing unit (GPU) is NVIDIA Tesla V100, and the computer language is Python 3.6. The model was implemented on Keras, and the proposed network was meant for ship object detection. For training, the Adam [24] gradient descent method, using the first-order moment estimation and second-order moment estimation of the gradient was used to adaptively adjust the learning step size of each parameter. The Adam attenuation coefficients are 0.9 and 0.999, respectively, and the batch size was set to 64. Each iteration (epoch) randomly arranges the dataset. The training termination condition is that the value of the loss function remains almost unchanged. FPN + VGG training takes 32 h in our GPU implementation, and 48 h with FPN+ ResNet-101.

### 4.2. Anchor Boxes Generation

Considering that the number of anchor boxes at each position should not be too large or too small, the number k of anchor boxes is selected to be 6, 9 and 12. The training method used by the FPN model is called alternative training. The detection algorithm model evaluates the overall detection accuracy of the test set of the dataset. Table 1 presents the comparison of test results with different models and different values of k. When k = 9, higher accuracy can be obtained in different network models. Figure 5a shows the convergence process of the loss function during the training of the FPN + VGG backbone network when k = 9, and the anchor box is the SSD-Kmeans cluster. In Figure 5b, the backbone network is changed to FPN + Resnet101. It is shown that the loss convergence speed of FPN+Resnet101 backbone network is slightly faster than the FPN + VGG network.

**Table 1.** Of different k values via shape clustering on different models of ship detection.

| SSD-Kmeans | Backbone | Accuracy (%) |
|:----------:|:--------:|:------------:|
| K = 6 | FPN + VGG | 95.6 |
| K = 9 | FPN + VGG | 96.675 |
| K = 12 | FPN + VGG | 96.3 |
| K = 6 | FPN + Resnet101 | 96 |
| K = 9 | FPN + Resnet101 | 97.3 |
| K = 12 | FPN + Resnet101 | 96.8 |

After training, the test dataset is used to evaluate the model in different scenarios. In the sea scene, the detection accuracy of different ship sizes was close to 100%, and no missed or false detection occurred. This shows that the model works well under no interference conditions. Near the islands and ports, ships of all sizes have high accuracy and no background implied that there are no false identifications. In the offshore area, due to the complicated background and low resolution, a few

items of background debris similar to the ship were misidentified as ships. As presented in Table 2, the accuracy of this algorithm was based on different backbone networks.

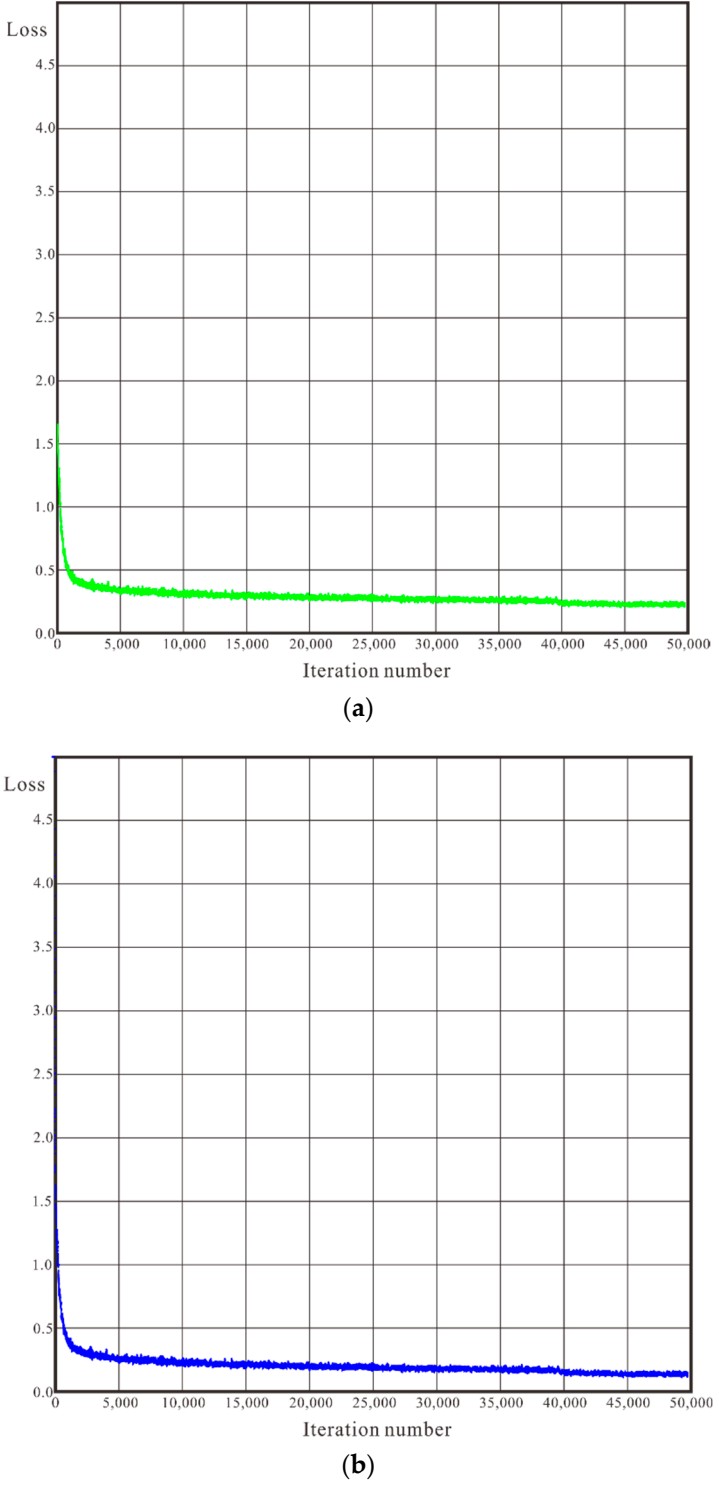

**Figure 5.** Loss function convergence. (**a**) SSD-Kmeans (k = 9) FPN based on VGG loss; (**b**) SSD-Kmeans (k = 9) FPN based on ResNet101 loss.

**Table 2.** Detection accuracy of the ship detection model SSD-Kmeans FPN utilizing different backbone networks (VGG and ResNet101) in four different scenarios (sea area, islands, harbor and offshore).

| Scenarios | SSD-Kmeans FPN + VGG | | SSD-Kmeans FPN + ResNet101 | |
|---|---|---|---|---|
| | Training Accuracy (%) | Validation Accuracy (%) | Training Accuracy (%) | Validation Accuracy (%) |
| Sea area | 100 | 99 | 100 | 99 |
| Islands | 100 | 96.3 | 100 | 97.6 |
| Harbor | 100 | 98.2 | 100 | 98.6 |
| Offshore | 100 | 93.2 | 100 | 94 |

*4.3. Analysis and Discussion*

After training, the test dataset is used to evaluate the model by comparing it with the RPN based on VGG/ResNet, FPN based on VGG/ResNet, Yolo, and SSD-Kmeans FPN models, and the evaluation indicators corresponding to each algorithm are calculated and categorized into the ship true detection rate $P_d$, the false detection rate $P_f$, and F1 Score defined as follows:

$$P_d = \frac{TP}{TP + FP} \tag{12}$$

$$P_f = \frac{FN}{TP + FN} \tag{13}$$

$$F_1 = \frac{2 \times P_d^* \times \left(1 - P_f\right)}{P_d + \left(1 - P_f\right)} \tag{14}$$

where *TP* is the number of true detections of ship objects, *FP* is refers to the unrecognized ship objects, *FN* is the number of false detections of ship objects and, therefore, *TP* + *FP* is the actual number of true ship detections and *TP* + *FN* denotes the total number of ship objects detections.

The test results are detailed in Table 3. The $P_d$ of the SSD-Kmeans FPN proposed in this paper is 98.62%, $P_j$ is 10.07% and F1 Score is 0.941. Specifically, as compared with the models based on RPNs, the detection accuracies of FPNs based on backbone networks increased by 5% and 4%. The results show that the FPN works better in small-object ship detection for SAR images. Furthermore, true detection accuracy of SSD-Kmeans FPN compared with Yolo improved by 12%, and the false detection rate reduced by 12%. The results show that the two-stage detection accuracy based on FPN is significantly higher than that of one-stage detection.

**Table 3.** Test results of the four models.

| Model | Backbone | Sea Area | | Islands | | Harbor | | Offshore | | Average | | |
|---|---|---|---|---|---|---|---|---|---|---|---|---|
| | | Pd (%) | Pf (%) | Pd (%) | Pf (%) | Pd (%) | Pf (%) | Pd (%) | Pf (%) | Pd (%) | Pf (%) | F1 Score |
| Yolo | —— | 92.82 | 12.92 | 92.32 | 14.07 | 83.46 | 29.19 | 78.21 | 32.84 | 86.71 | 22.25 | 0.819 |
| RPN | VGG | 95.9 | 6.25 | 93.75 | 10.05 | 89.28 | 23.31 | 79.47 | 28.67 | 89.6 | 17.07 | 0.861 |
| FPN | VGG | 96.53 | 4.03 | 95.56 | 4.42 | 91.07 | 19.93 | 90.04 | 22.58 | 93.3 | 12.74 | 0.901 |
| SSD-kmeans + FPN | VGG | 99.1 | 3.63 | 98.32 | 3.56 | 97.31 | 14.01 | 97.27 | 19.92 | 98.0 | 10.28 | 0.936 |
| RPN | Resnet101 | 96.4 | 6.25 | 94.26 | 6.28 | 92.85 | 22.07 | 85.73 | 29.68 | 92.31 | 16.07 | 0.879 |
| FPN | Resnet101 | 97.82 | 3.47 | 97.61 | 3.7 | 97.72 | 19.32 | 93.65 | 22.31 | 96.7 | 12.2 | 0.92 |
| SSD-Kmeans + FPN | Resnet101 | 99.2 | 3.32 | 98.9 | 3.51 | 98.85 | 13.56 | 97.53 | 19.89 | 98.62 | 10.07 | 0.941 |

Therefore, the algorithm proposed in this paper has better accuracy than the traditional CNN models in detecting small-object ships in complex scenarios. The FPN based on VGG/ResNet and SSD-Kmeans FPN both use FPNs embedded in RPNs, and due to the introduction of high-resolution features mapping, these models are more conducive to small-object location and recognition. The anchor

boxes based on the shape clustering algorithm provide more accurate proposals and, therefore, the accuracy of object positioning is improved. Furthermore, in complex scenarios such as islands, ports, and offshore buildings, the detections are prone to errors, and in such cases, two-stage models are significantly better than one-stage detection models.

The detection results obtained in different complex scenarios are presented in Figure 6. It can be seen that all the four models demonstrated effective performance when the background was an ocean; and, when the ship approached offshore lands, islands, and ports, although the models were still able to detect the ship, false positives or underreports were generated. Underreporting occurs mainly because the RPN extracts features from the last convolution layer, the relative positional deviation of which will be large in the case of small-object detection. The FPN-based algorithm can improve the accuracy of small-object positioning. False reporting is the misidentification of machineries on buildings and ports as ships. As can also been seen from Figure 6, the false reporting rates of the four models are similar. The underreporting rate of the proposed algorithm is low; however, its overall detection result is the best among all the four algorithms.

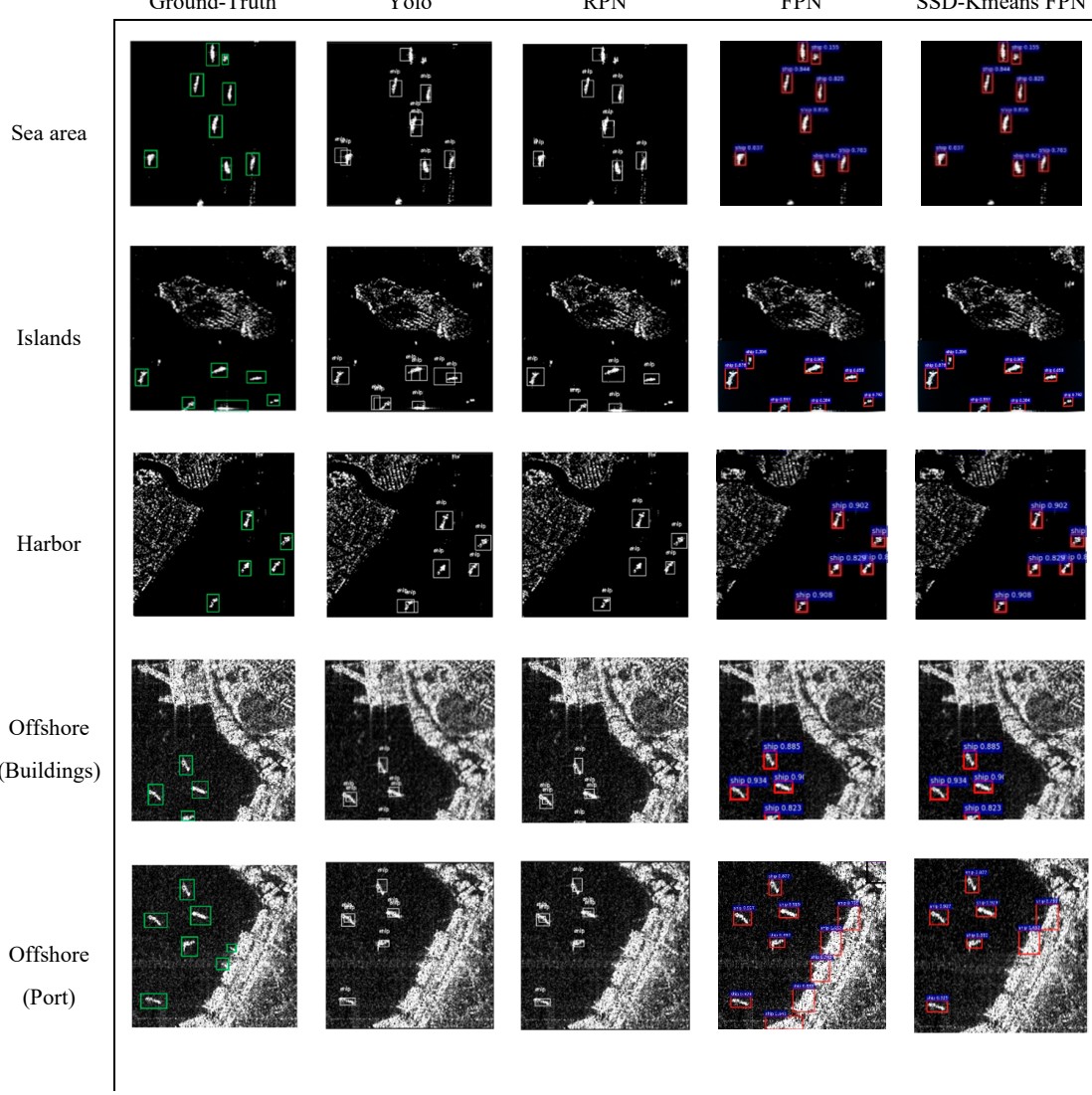

**Figure 6.** Comparison of ship object-detection results of four models with the ground-truth in complex scenarios. Each row represents a scene (sea area, islands, harbor, and offshore), and each column corresponds to a different model (Yolo, RPN, FPN, SSD-Kmeans FPN).

## 5. Conclusions

We presented an FPN model using anchor boxes obtained through SSD-Kmeans clustering for small-object ship recognition in complex backgrounds. Different scenarios of Sentinel-1 SAR images were used to verify the proposed model. The experimental results showed that the optimized anchor boxes FPN network has a detection accuracy rate close to 100% for different scales in the case of small target objects (with a scale below 64 pixels). Compared with the two-stage object-detection methods; FPN, faster R-CNN, and one-stage detection method, Yolo, in complex scenarios, the detection accuracy was demonstrated to improve with the use of the proposed method, and the false detection rate and the omission ratio of the target ship also reduced. The proposed model is also suitable for multi-scale and multi-target recognition in simple scenarios. The model also has several advantages over the existing models for detection of small target ships in complex scenarios.

**Author Contributions:** P.C. conceived and designed the algorithm and contributed to the manuscript and experiments; H.Z. was responsible for the construction of ship detection dataset, the outline for the manuscript and the first draft of the manuscript; Y.L. and B.L. supervised the experiments and also contributed to the construction of the dataset; P.L. performed oil spill detection using machine learning. All authors have read and agreed to the published version of the manuscript.

**Funding:** This research was funded by the National Natural Science Foundation of China, grant number 51609032.

**Conflicts of Interest:** The authors declare no conflict of interest.

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
