# Peer review of "Detection of Small Ship Objects Using Anchor Boxes Cluster and Feature Pyramid Network Model for SAR Imagery"

_jmse, doi:10.3390/jmse8020112_

Round 1
Reviewer 1 Report
Interesting research methodology design. The research topic is useful, objectives are clear and were met by the authors.
Author Response
Thank you for your review, and I will amend the related questions in this article.
Reviewer 2 Report
How about considering to change the title
"Using Optimized FPN Model for Detection of Small Ship Objects"
to
"(A) Detection Technique for Small Ship Objects using Optimized FPN Model".
96 2. Related works
Author Response
Thank you for your review, your suggestions have helped me a lot and I will take your suggestions.
Reviewer 3 Report
This manuscript considered satellite-SAR image-processing for small ship detection using Feature Pyramid Network and k-means clustering. The work seems to conduct some sorts of novelty but the following comments need to be addressed.
1) The abbreviation FPN in the title is not widely common and not uniquely defined in the literature. FPN can mean (fixed pattern noise), (fluorescent polymeric nanoparticles), or (fuzzy polynomial neurons).
2) It is important to define the abbreviation, changing the the title to “Optimized Feature Pyramid Network Model for Small-Ships Object Detection” for example.
3) A literature review of closely related works using FPN is missing.
4) There are some minor grammatical and technical language errors that need correction. For example (but not limited to): “problems in object detection” to “ problems of object detection” or “ object detection problems”; “Proposal Method” to “ Proposed method”; “algorithm is a clustering based on” to “is a clustering algorithm based on”; and “Calculate Eq. (1) and (2) repeatedly until” to something else different from an instruction/order.
5) The symbols before Eq(1) are defined through a different font size or lines spacing and inconsistent with the remaining text. The same observation through the entire text. Also, the symbol x in Eq(1) is different from the preceding line when it was introduced. What is xi?
6) K-means algorithm is very famous in the literature and it is not the authors original contribution. But if the authors choose to re-introduce it, this would better be within the context of this work but not in the very original form of the algorithm. For example, what does the sample x represent in this framework?. Also, the algorithm must not be introduce through an imperative form (Calculate …, Obtain …, you need to).
7) “Ground truth” is not defined in framework of this manuscript, and its introduction in the paragraph before Eq(4) is confusing. The same for “Anchor box”. Why both are capital-letters in the middle of phrases?. In fact the paragraph before Eq(4) is not clear at all.
8) Description of sized and dimensions of training and testing data-sets and the differences between is missing.
Author Response
Thank you for your review, your suggestions have helped me a lot and I will take your suggestions.
Response to Reviewer 3 Comments
Point 1: The abbreviation FPN in the title is not widely common and not uniquely defined in the literature. FPN can mean (fixed pattern noise), (fluorescent polymeric nanoparticles), or (fuzzy polynomial neurons).

Response 1: Feature Pyramid Network (FPN) is embedded in the RPN network, and each layer is independently predicted.
Point 2: It is important to define the abbreviation, changing the title to “Optimized Feature Pyramid Network Model for Small-Ships Object Detection” for example.
Response 2: The related description of FPN in Section 2.2, and related improvements to FPN in Section 3.1. Thanks for the comments on the title, we will modify it further.
Point 3: A literature review of closely related works using FPN is missing.
Response 3: The relevant references are cited in the introduction and section 2.2. Add two new literature descriptions as suggested. For “M2Det: A Single-Shot Object Detector based on Multi-Level Feature Pyramid Network”, this literature add the mutil-level information to the mutil-scale of the FPN structure and design the MLFPN network structure. Combine the same scale in different levels to improve detection accuracy.
Point 4: There are some minor grammatical and technical language errors that need correction. For example (but not limited to): “problems in object detection” to “ problems of object detection” or “ object detection problems”; “Proposal Method” to “ Proposed method”; “algorithm is a clustering based on” to “is a clustering algorithm based on”; and “Calculate Eq. (1) and (2) repeatedly until” to something else different from an instruction/order.
Response 4: The next step is to further polish the language.
Point 5: The symbols before Eq(1) are defined through a different font size or lines spacing and inconsistent with the remaining text. The same observation through the entire text. Also, the symbol x in Eq(1) is different from the preceding line when it was introduced. What is xi?
Response 5: xi is the target ground-truth for each ship. And it will be modified in the paper.
Point 6: K-means algorithm is very famous in the literature and it is not the authors original contribution. But if the authors choose to re-introduce it, this would better be within the context of this work but not in the very original form of the algorithm. For example, what does the sample x represent in this framework?. Also, the algorithm must not be introduce through an imperative form (Calculate …, Obtain …, you need to).
Response 6: The shape and size of Anchor boxes are a set of hyperparameters. In the actual SAR image, the size of the target changes a lot. Using the mechanism of traditional FPN will cause slower convergence of border regression. Therefore, consider clustering to select the initial Anchor boxes that are more in line with the ship shape features in the SAR image.
The k-means algorithm proposed by J Mac Queen is a clustering algorithm based on the distance evaluation scale to the prototype. Due to its simple and efficient characteristics, it is widely used in clustering problems. In this paper, the k-means algorithm based on the shape similar distance (SSD) clustering is used to obtain k initial anchor boxes, namely SSD-Kmeans.
Detailed content will be revised in the paper.
Point 7: “Ground truth” is not defined in framework of this manuscript, and its introduction in the paragraph before Eq(4) is confusing. The same for “Anchor box”. Why both are capital-letters in the middle of phrases?. In fact the paragraph before Eq(4) is not clear at all.
Response 7: There is a description of Ground-Truth and Anchor box in 2.2, but it is not detailed enough, so section 2.2 is amended to “Four rectangular boxes(32pixels*32pixels,64pixels*64pixels,128pixels*128 pixels) with different pixel areas are allocated on the five feature mapping layers(P1,P2,P3,P4,P5), and multiple aspect ratios(1:2,1:1,2:1) are used at the same time. These rectangular boxes are called Anchor boxes. On each feature mapping layer, sliding each window with Anchor boxes as a fixed area to generate a large number of proposals. Then calculate the intersection ratio (IOU) of the proposals and the target truth box (Ground truth), choose the closest to Ground truth as the ROI.”
Point 8: Description of sized and dimensions of training and testing data-sets and the differences between is missing.
Response 8: There are 43,819 data sets in total, and the training set, validation set, and test set are divided by 70%, 20%, and 10%. The image size in the dataset is 256*256 pixels